# Chromosomal Instability, Selection and Competition: Factors That Shape the Level of Karyotype Intra-Tumor Heterogeneity

**DOI:** 10.3390/cancers14204986

**Published:** 2022-10-12

**Authors:** Tom van den Bosch, Sarah Derks, Daniël M. Miedema

**Affiliations:** 1Laboratory for Experimental Oncology and Radiobiology, Center for Experimental and Molecular Medicine, Cancer Center Amsterdam and Amsterdam Gastroenterology Endocrinology Metabolism, Amsterdam University Medical Centers—Location AMC, 1105 AZ Amsterdam, The Netherlands; 2Oncode Institute, 1105 AZ Amsterdam, The Netherlands; 3Department of Medical Oncology, Amsterdam University Medical Centers—Location VUmc, 1081 HV Amsterdam, The Netherlands

**Keywords:** chromosomal instability, intra-tumor heterogeneity, aneuploidy, immunological pressure

## Abstract

**Simple Summary:**

Each cancer consists of billions of cells. These cells are far from identical; hence, the population of cells that constitute a tumor is heterogeneous. A salient property that varies between cells in a tumor is their karyotype, the number and configuration of the chromosomes. The level of karyotype heterogeneity can be used to predict the survival of a patient. In this review, we describe the processes that shape the level of karyotype heterogeneity in a cancer.

**Abstract:**

Intra-tumor heterogeneity (ITH) is a pan-cancer predictor of survival, with high ITH being correlated to a dismal prognosis. The level of ITH is, hence, a clinically relevant characteristic of a malignancy. ITH of karyotypes is driven by chromosomal instability (CIN). However, not all new karyotypes generated by CIN are viable or competitive, which limits the amount of ITH. Here, we review the cellular processes and ecological properties that determine karyotype ITH. We propose a framework to understand karyotype ITH, in which cells with new karyotypes emerge through CIN, are selected by cell intrinsic and cell extrinsic selective pressures, and propagate through a cancer in competition with other malignant cells. We further discuss how CIN modulates the cell phenotype and immune microenvironment, and the implications this has for the subsequent selection of karyotypes. Together, we aim to provide a comprehensive overview of the biological processes that shape the level of karyotype heterogeneity.

## 1. Introduction

Intra-tumor heterogeneity (ITH), defined as diversity in the cell populations that form a tumor, is one of the hallmarks of cancer [1]. The amount of ITH varies strongly between individual cancers and appears to be a crucial determinant of patient survival [2,3,4,5]. According to the Darwinian theory of evolution, genetic variation in populations, and ITH in cancers, fuels evolution [6]. Indeed, mounting evidence suggests that ITH is associated with tumor evolution, a propensity to metastasize, resistance to therapy and overall, a dismal prognosis [7,8,9,10,11,12,13,14].

The observation that cancers have abnormal karyotypes and the possible consequences for cancer progression were reported more than a hundred years ago [15]. The fact that karyotypes can vary substantially between cancers was recognized by Peter Nowell in 1976, when he rewrote the seminal paper of clonal heterogeneity and evolution in cancer [16]. In the 1980’s, the impact of karyotype ITH on chemotherapy-resistance and tumor progression was described [17,18]. Technological developments during the last few decades have allowed large-scale analyses to address ITH in many cancers of different types. Projects such as The Cancer Genome Atlas (TCGA) and the International Cancer Genome Consortium (ICGC) allowed detailed studies of how ITH in karyotypes impacts the biology of cancer using next-generation sequencing. Taylor et al. analyzed 10,522 cancer genomes and observed aneuploidy in 88% of cancers [19]. The level of aneuploidy was found to be associated with survival and metastases across cancer types [20,21]. Van Dijk et al. reported that although aneuploidy is a prognostic factor, ITH in copy numbers is a better predictor of patient survival than aneuploidy itself [4].

More than a century of cancer research has, thus, unequivocally demonstrated the importance of karyotype ITH for cancer evolution and progression. ITH likely increases the fitness of cancers by a numbers game; each cell is under selective pressure; therefore, the chance that a cancer progresses increases with ITH. Therefore, a pivotal question that emerges is as follows: what determines the level of karyotype ITH? Further building onto Darwin’s framework of evolution, early reports on ITH argued that the level of karyotype ITH results from a balance between genomic instability and subsequent selection [16,18,22].

Genomic instability is the fundamental process that drives ITH [23,24,25]. Microsatellite instability (MSI) results in a high tumor mutational burden (TMB) and ITH at the single-nucleotide level. Chromosomal instability (CIN) is the equivalent process at a large scale, affecting thousands to hundreds of millions of base pairs simultaneously. Through mis-segregation of chromosome (parts), CIN results in aneuploidy and karyotype ITH. Studies based on multi-region bulk sequencing [26,27,28], live imaging of cell divisions in organoids [29], single-cell karyotyping [30,31] and machine learning applied to digital pathology [32] have recently demonstrated that ongoing CIN is a fundamental characteristic of cancer. Yet, despite ongoing CIN, a dominant karyotype can be identified in almost all malignancies from bulk samples [33,34,35]. It follows that selective forces must be at play to counter the effect of CIN and reduce the level of karyotype ITH to fluctuations around one or a few selected karyotype(s) (clones).

Here, we aim to provide a comprehensive overview of the factors that impact the level of karyotype ITH (Figure 1). In Section 2, we describe how different types of CIN create a spectrum of karyotypes. Next, we describe how cells with different karyotypes undergo selection by cell intrinsic limitations to cell viability and fitness in Section 3, and by pressure from the microenvironment exerted by, e.g., tumor immune infiltrate in Section 4. In Section 5, we review how competition between cells, also in the case of equal fitness, impacts karyotype ITH.

## 2. Numerical and Structural CIN Generates New Karyotypes

The following two categories of CIN are typically distinguished: numerical CIN and structural CIN. CIN that alters the number of copies of entire chromosomes is defined as numerical CIN; CIN that alters the nucleotide sequence of large parts of chromosomes, e.g., through translocations or gains/losses of sub-chromosomal regions, is defined as structural-CIN [36]. Numerical CIN, hence, does not alter the nucleotide sequence in any chromosome, but only the number of copies of chromosomes, resulting in a state of aneuploidy. Numerical CIN emerges mostly from segregation errors during mitosis, with the underlying causes including mitotic checkpoint defects, centrosome amplification and chromosome oscillation [37,38,39]. Perhaps the most striking event that induces CIN is polyploidization or genome doubling, caused by cell fusion, endoreplication, or histone stress [40,41]. Genome doubling results in genomic instability [42] and is, in particular, associated with numerical CIN [43]. Genetic defects associated with numerical CIN include the following: overexpression of *AURKA*, linked to centrosome amplification, depletion of *CENP-E*, resulting in checkpoint defects, and mutations in *BRCA1/2*, associated with overall dysregulation of cell-cycle proteins [38].

Structural CIN generates aneuploidy through focal amplification or loss of parts of chromosomes, disintegration of chromosome parts into extra-chromosomal structures, such as double minutes, and complex rearrangements of large nucleotide sequences in, e.g., chromothripsis [44]. Causes linked to structural CIN include defects in the machinery to repair double strand breaks and replication stress [45]. Other mechanisms linked to structural CIN are incomplete DNA decatenation, telomere dysfunction, non-allelic homologous recombination and unequal crossing over during meiosis [46]. Structural CIN is further thought to be impacted by the nuclear architecture of the cell [47]. In some cancers, telomere length and enhanced telomere attrition are also associated with increased structural CIN [48].

Although structural CIN and numerical CIN have been associated with different phenotypes [49], they tend to co-occur in the majority of cancers [50,51]. Deconvolution of karyotypes measured from cancer genomes might help to identify the underlying causes and interactions between different types of CIN. Deconvolution of aggregated genomic aberrations was first introduced by Alexandrov and co-workers to find signatures in single-nucleotide mutations and to study the etiology of mutational processes [52,53]. Recently, analogous approaches of deconvolution have been applied to large datasets of genome-wide chromosomal copy numbers to characterize the patterns of copy number variations (CNVs) [44,54,55,56,57]. These studies linked specific CNV signatures to the underlying biological processes, such as homologous recombination deficiency. Interestingly, the etiology of some CNV signatures could not yet be explained, suggesting unknown biological process that drive distinct types of CIN. From the current perspective of ITH that we take in this review, we note that, regardless of the type of CIN, any new karyotype introduced can increase karyotype ITH.

## 3. Cell intrinsic Selection for Karyotypes

### 3.1. Aneuploidy Tolerance

Ongoing CIN continuously generates cells with new karyotypes. Each cell is constantly subject to selective pressures and if a few or none of the cells with a new aberrant karyotype is able to survive, karyotype ITH will be low. Indeed, the first and most eminent requirement for the presence of karyotype ITH is that cells in a cancer can cope with aneuploidy. Non-malignant cells, in general, cannot cope with aneuploidy. For instance, when aneuploidy is induced in non-malignant cells, this is immediately associated with reduced cell proliferation rates [58,59].

The detrimental effect aneuploidy can have on cells and organisms becomes strikingly clear during embryonic development [60]. Monosomy and the large majority of trisomies of autosomal chromosomes are lethal in the embryonic stage. Trisomy of chromosomes 21 (Down), 13 (Patau) and 18 (Edwards) are the only trisomies that are compatible with post-natal life, but all are associated with severe physical and cognitive disorders [61]. Sub-chromosomal CNVs are more common in the human population, but also negatively associated with traits such as adiposity, liver damage and decreased intelligence, as was found by a large study that analyzed data from the UK Biobank [62]. Of note, the detrimental effect of aneuploidy on development is not limited to humans and has also been observed in, e.g., mice [63] and maize [64].

While aneuploidy is largely absent in healthy tissue, the majority of cancers are aneuploid [19,33]. Hence, there appears to be an aneuploidy paradox, with CNVs widely observed in malignancies derived from virtually all tissues, but associated with impaired survival in healthy cells [65]. A possible explanation of this apparent paradox is that healthy cells have intact regulatory mechanisms that lead to cell cycle arrest in case of aneuploidy, the mechanisms that are deactivated in the transformation of healthy tissue into a malignancy [66,67].

One gene involved in these regulatory mechanisms is *TP53*, the guardian of genome integrity and the most frequently mutated gene across cancers [68]. Cancers with clonal mutations in *TP53* are enriched in CNVs [19]. Cancers with no mutation in the *TP53* gene often have other mutations in the *TP53* pathway, e.g., in *CDKN2A* [69]. However, a recent study involving organoid cultures of *TP53* deficient cells questioned the universality of *TP53* to protect against aneuploidy [70]. Moreover, other genes, including *BCL9L* and *MAPK14,* have been implicated in suppressing tolerance to aneuploidy [71,72]. Together, these studies suggest that through genetic alterations, malignant cells become tolerant to aneuploidy, which opens the door to karyotype ITH in a cancer. In addition, epigenetic differences might also contribute to the tolerance of ITH. Cancer stem cells, for example, differ only epigenetically from more differentiated cancer cells, and are thought to have low, or at least lower, levels of CIN compared to the latter [73,74].

### 3.2. Aneuploidy and Cell Fitness

The question of how aneuploidy impacts the phenotype and fitness of cells emerges. The most likely mechanism is the direct effect aneuploidy has on gene expression [67]. Chromosomes contain an unequal number of genes and aneuploidy affects different chromosomes; hence, it also impacts a different number of genes. Interestingly, trisomies of chromosomes 13, 18 and 21 are compatible with post-natal life and these chromosomes contain the lowest number of genes of all autosomal chromosomes, with only ~500 genes located on chromosome 21 for instance [75]. In addition, in cancer, small chromosomes are more frequently lost compared to large chromosomes [76]. This suggests that, overall, the effect of a CNV on phenotype depends on the number of genes that are involved.

The relation between gene and protein expression on the one hand, with the number of gene copies on the other, has been widely demonstrated [77,78,79]. In fact, CNVs appear to be the primary driver of variations in gene expression in cancers [80]. The impact of aneuploidy on gene expression is also evident from the observation that the reverse problem can be solved; copy number profiles can be inferred from measurements of gene expression of bulk samples [81] and single-cells [82,83], which is only possible when CNVs significantly impact gene expression. Hence, despite the activation of several feedback mechanisms to compensate against altered gene expression through copy number changes [84,85,86], CNVs impact cell phenotype directly through changing the number of gene copies. This might explain why homozygous deletions of large DNA segments are rarely found in cancers [87,88], because homozygous deletion erases genes from a cell, which could inflict a significant fitness penalty to a cell. Of note, it might be the case that although CNVs impact the overall gene expression, the gains and losses selected for in cancer emerge because they contain oncogenes and tumor suppressors, respectively [89,90]. Moreover, aneuploidy-induced changes in gene and protein expression can result in a state of cellular stress [91].

The gene expression profile of a cell, hence, depends on its karyotype. It follows immediately that karyotype ITH results in gene expression ITH. As gene expression overall has a strong correlation to protein expression [92], karyotype ITH might translate to phenotypic ITH.

### 3.3. Tissue Specificity of CNVs

The altered gene expression of a new karyotype impacts cell fitness in a context-specific manner [93]. It is against the background of the gene expression profile present in a cell before a CNV occurs that the impact on cell fitness of that particular CNV is determined. Cells from different tissues have different phenotypes and gene expression patterns. Indeed, the patterns of aneuploidy observed in cancers are also highly tissue-specific [94]. The pattern of CNVs is sufficiently tissue-specific that it can be used to identify the tissue of origin of a cancer [95]. Moreover, CNV patterns show large overlap within, e.g., gastrointestinal cancers or gynecological cancers, but are very different from the CNVs observed in cancer from distinct tissues, such as lung or breast cancer [19].

Various mechanisms have been proposed for how tissue specificity of aneuploidy emerges through the effect on gene expression. The existence of hard wiring of gene expression in healthy tissue was recently proposed as a mechanism of tissue specificity in aneuploidy [96]. The hypothesis of hard wiring of gene expression from the cell of origin is supported by the overall similarity of CNVs in the primary tumor and distant metastasis [97,98]. Alternatively, a subset of genes could be the target through which aneuploidy acts, e.g., by copy number changes in genes associated with cell proliferation [99], or gains of oncogenes and loss of tumor suppressor genes [90]. In particular, amplification of small genomic regions through extra-chromosomal DNA appears to impact oncogenes [100]. The number of distinct karyotypes that result in a fit and viable phenotype is, thus, limited by the tissue of origin for each cancer. Importantly, this restricts the level of karyotype ITH, as the majority of new karyotypes will result in gene expression changes that are not compatible within the background of tissue-specific gene expression.

### 3.4. Divergence of CNVs during Carcinogenesis

Within tumors derived from the same tissue, substantial inter-patient heterogeneity in aneuploidies nevertheless exists [101]. Patterns of aneuploidy coalesce with, or perhaps shape, subtypes of breast cancer and glioblastoma [19]. In colorectal cancer, mucinous tumors and non-mucinous tumors have distinct patterns of aneuploidy [102]. Furthermore, regardless of the tissue of origin, MSI tumors often have a much lower aneuploidy load compared to microsatellite stable tumors of the same tissue of origin.

Specific mutations that accumulate during tumor evolution further shape the malignant cell karyotype. This can be an effect of a single (onco) gene that is either mutated or amplified. For example, *KRAS* is either mutated or amplified in colorectal cancer [103]. A mutated *KRAS,* hence, is negatively associated with an amplification of *KRAS* in colorectal cancer. Similarly, loss of heterozygosity (LOH) after an oncogenic hit in a tumor suppressor gene is a common mechanism that eliminates the remaining wild-type allele of a tumor suppressor [88,104]. Alternatively, mutations can impact CNVs of other genes, and. e.g., manifest as different routes to genomic instability, such as those observed in breast cancer for *BRCA1/2* mutations and *CCNE1* amplification [105]. In head and neck cancers, mutations in *TP53* co-occur with a loss of chromosome arm 3p, which is associated with poor survival [106].

Co-occurring CNVs similarly suggest that one copy-number event impacts the probability for the next event [107]. Such interactions between CNVs have been reported for brain tumors [108] and ovarian serous carcinomas [109]. Recent work by Kester et al., using serial passaging of colorectal cancer-derived organoids, revealed that loss of chromosome 4 is observed after loss of chromosome 18, suggesting that chromosome 18 alters the fitness landscape and evolutionary trajectory in carcinogenesis [110].

Early driver mutations in tumorigenesis can also shape the genome-wide spectrum of CNVs. *BRCA* mutations in breast cancer are associated with a particular pattern of CNVs [111]. Similarly, mutations in both alleles of *Trp53* shape the trajectory of tumor evolution, resulting in recurring patterns of aneuploidy in a mouse model of pancreatic ductal adenocarcinoma [112]. Moreover, whole-genome doubling increases CIN and, in particular, results in whole chromosome CNVs [43].

In addition, epi-genetic changes alter malignant cell phenotypes and gene expression profiles and can, hence, translate into specific patterns of aneuploidy [113]. Together, a picture emerges, where the spectrum of available karyotypes co-evolves with the cell phenotype that develops before, as well as after, malignant transformation. The accumulated genetic changes that shape the (karyotype) fitness landscape are analogous to the accumulated epi-genetic changes that determine the fitness landscape of differentiating cells, as proposed by Waddington [114]. The smaller the spectrum of fit karyotypes, the lower the level of karyotype ITH that might be expected in a malignancy. From this perspective, events of punctuated evolution, such as chromothripsis and whole genome doubling, can be viewed as radical and sudden jumps within the karyotype fitness landscape that (temporally) result in high karyotype ITH [30,115,116] (Figure 2).

## 4. Selection for Karyotypes by the Microenvironment

Selection acting on karyotypes is not solely a cell intrinsic process. Tumors are increasingly understood as complex ecological systems, consisting of a stromal compartment, vasculature and an immune infiltrate that interact with the malignant cells and shape evolution [117,118]. In healthy tissue, cell proliferation and death are under the strict control of the microenvironment. A key function of early driver mutations in cancer formation is to render cells less dependent on cues from the microenvironment [117,119].

In established cancers, an interaction between malignant cells and the microenvironment persists. Upon xenotransplantation or in vitro, cancer cells grow in strikingly different environments than in the primary tumor from which they were derived. Multiple studies reported karyotype evolution upon xenotransplantation of carcinomas [120,121,122,123]. With each successive passage in mice, the karyotype diverges further from the original tumor [124]. Ben-David et al. studied cell lines under various culture conditions, and found that clonal divergence was dictated by culture condition-specific selection, rather than a stochastic process [125]. Moreover, the recent studies by Ippolito et al. and Lukow et al. demonstrate that karyotype adaptation is a strategy used to handle changing culture conditions imposed by drug therapies [126,127].

### 4.1. Immunoediting

For in situ tumors, the immune system exerts selective pressure on malignant cells [128]. Surveillance of CRC metastasis over time demonstrated immune escape mechanisms, specifically observed in aneuploid tumors [129]. Clones with a high level of immunogenicity underwent early elimination before dissemination took place, while persisting and progressing clones acquired immune escape mechanisms [129]. A similar observation was reported in in vivo mouse studies, where injection of heterogeneous cell populations in immunocompetent mice led to the rejection of immunogenic tumor clones, which resulted in less heterogeneity [130]. Together, these studies show that immune pressure reduces ITH.

A central tenet of immunoediting is the presentation of antigens by malignant cells. The presentation of antigens by malignant cells attracts immune cells into the tumor microenvironment that, in turn, target the malignant cells. MSI tumors, which have many neoantigens due to high mutation rates, are characterized by strong immune infiltrates and a relatively good prognosis [131]. In lung cancers, which are typically microsatellite stable, chromosome loss is a mechanism that reduces neoantigen load and avoids immunosurveillance [132]. Importantly, the mechanism of reducing neoantigen load through chromosome losses might contribute to acquired resistance upon treatment with immune checkpoint inhibitors [133]. Neoantigens are presented to the immune system through human leukocyte antigens (HLA). Loss of heterozygosity of HLA has been described as an immune escape mechanism in later stages of lung cancer evolution [134]. Another mechanism by which the presentation of neoantigens is reduced in aneuploid cancers is through the high flux unstable wild-type proteins that have a competitive advantage over neoantigens in binding to major histocompatibility complexes (MHC) for antigen presentation [49].

Of note, the immune infiltrate of tumors shows substantial spatial variations [135,136,137]. Hence, immune ITH might be directly linked to karyotype ITH [132].

### 4.2. CIN Modulates the Tumor Immune Microenvironment

We reviewed above how the tumor-immune infiltrate selects for specific karyotypes. The tumor immune microenvironment itself, however, is partially shaped by CIN and aneuploidy. For example, in gastroesophageal cancers, tumors with high levels of CIN and low CD8 T cell scores showed enrichment of cell cycle pathways, such as *MYC* pathway expression and cyclin E1 (*CCNE1*) gene amplification [138]. The association between *MYC* and immune suppression has been described in a variety of cancers [139]. In immune-competent lymphoma, pancreatic cancer and lung cancer mouse models for instance, *MYC* activation promotes the influx of macrophages and neutrophils and promotes the loss of T cells, B cells, and NK cells [140]. *MYC* overexpression also induces angiogenesis through the release of candidate immunomodulatory factors, including *CCL9*, *IL-23* and *VEGFA* [141]. *MYC* also modulates the expression of immune checkpoint molecules *PD-L1* [142]. Thereby, oncogenic drivers amplified through CIN can promote immune suppression.

The immediate consequence of chromosomal mis-segregation is the induction of a pro-inflammatory response to ensure elimination of cells with an unbalanced karyotype by the immune system [143]. Aneuploid cells, for instance, upregulate surface proteins, such as the NKG2D ligand family, that trigger recognition by the innate immune cells, such as natural killer (NK) cells. Upregulation of interleukin-6 (*IL-6*), *IL-8*, and *CCL2* in aneuploid tumors has also been described. One of the best described proinflammatory pathways is the cGAS-STING pathway [143,144].

The cGAS-STING pathway is an important initiator of an innate immune response and becomes activated when double strand DNA leaks into the cytosol, which is often observed in chromosomal instable cancers. As no free DNA should be in the cytosol, dsDNA are immediately recognized by receptors such as the DNA sensing molecule cyclic GMP-AMP synthase (cGAS), which will bind to the ER-membrane activator and stimulator of interferon genes (STING) [24]. After sensing dsDNA in the cytosol, STING stimulates the expression of type 1 interferons (IFNs), which promotes inflammation.

Interestingly, it has been shown that this inflammatory response can eventually become suppressed in CIN cancer cells due to loss of chromosome 9p that harbors the IFN gene cluster [24] or by upregulation of the non-canonical NF-kB pathway, which can inhibit the expression of type I IFNs [145]. Thereby, chronic STING stimulation, caused by the presence of CIN and type I IFN, will eventually lead to immune suppression [146].

### 4.3. Hypoxia

Fast growing tumors can have poor vascularization, which results in hypoxia. Hypoxia has been linked to genomic instability in general and chromosomal instability in particular [147,148]. A pan-cancer multi-omics analysis indicated that reprogramming of metabolism in hypoxic tumors might be driven by CNVs in hypoxia-associated genes [149]. In an experimental setting that involved in vitro cultures of cell lines, it was found that hypoxia induces copy number alterations [150]. Both these studies of Haider et al. and Black et al. suggested a direct link between unequal distributions of hypoxia and karyotype heterogeneity in tumors [149,150]. A recent study by Zhao et al. corroborated this observation and concluded that in the hypoxic tumor core, tumor evolution is accelerated through CNVs [151]. Similar to the effect of hypoxia, Graham et al. observed that glycolytic metabolic stress exerts selective pressure on the tumor karyotype, resulting in specific CNV signatures [152].

Selection from the microenvironment, through immunosurveillance, hypoxia or other mechanisms, acts to limit the number of fit karyotypes. The number of fit karyotypes a malignant cell can possess is, thus, determined by cell intrinsic and cell extrinsic factors, and dictates the level of karyotype ITH that can be obtained by a cancer.

## 5. Competition between Malignant Cells

### 5.1. Cell Competition and Relative Fitness

A new karyotype generated by CIN can result in a lower cell division time, and thereby an increased fitness compared to surrounding malignant cells [153]. Similarly, a karyotype that provides a reduced chance of cell death also provides a competitive benefit to expand within a malignancy [29]. Importantly, the relevant fitness of a malignant cell is defined relative to the fitness of other (malignant) cells. The increase is first compared to healthy cells and later compared to other transformed cells in the process of cancer development [154,155].

Therefore, malignant cells can also increase their net fitness by reducing the fitness of surrounding cells. This process of manipulating the fitness of surrounding cells was first described in *Drosophilla Melanogaster,* where the mutated *MYC* oncogene altered the fitness of surrounding cells [156,157]. Di Giacomo et al. found that also in cancer cells, *MYC* activation is a process that induces cell deaths in the surrounding cells and can facilitate the invasion of surrounding tissue [158]. Importantly, *MYC* is often amplified in cancer and the mechanism of *MYC* over-expression in Drosophila is amplification [159].

Recently, it was found that manipulation of fitness in surrounding cells might be a generic mechanism employed by cells at the onset of colorectal cancer [160,161,162]. Employing lineage tracing in the mouse intestine, Yum et al. observed that mutant *KRAS* and *PI3K* cells reduced the fitness of neighboring stem cells [160], while Van Neerven et al. and Flanagan et al. reported a similar effect in mutant *APC* cells [161,162]. Hence, negatively impacting the fitness of surrounding cells might be a key mechanism on which cancer-associated genes rely [163].

The interaction between malignant cells is not only one of negative interference, as clonal cooperation in cancer has also been described in the seeding of polyclonal tumors [164]. The cooperation between malignant cells might work indirectly through the microenvironment [165] or directly via autocrine/paracrine signaling [164]. Autocrine [166] and paracrine [167] signaling to co-stimulate growth of malignant cells in vitro have recently been reported, and could explain the sub-exponential expansion of cells seeded at low density [168].

Hence, signaling between malignant cells that positively or negatively impacts fitness can increase or decrease ITH, respectively. However, also in the case of equal and independent cell fitness, malignant cells interact with one another in the competition for space and resources.

### 5.2. Genetic Drift and Spatial Constraints

Genomic aberrations that alter the fitness of cells and drive cancer have been extensively catalogued [169,170]. The majority of mutations, however, appear not to alter the fitness of malignant cells [171]. In absence of genomic drivers, cancers evolve neutrally [172], resulting in high genetic diversity [173]. Indeed, in the absence of any fitness difference between malignant cells, all clones have equal chances to persist, and ITH obtains the maximum value [174].

Under neutral dynamics, there is, nevertheless, competition between malignant cells for nutrients and space. Due to random cell division and death rates, allele frequencies vary stochastically, a mechanism known as genetic drift [175,176]. Importantly, for any non-zero death rate, new karyotypes might be eliminated from the malignant cell population by chance alone through genetic drift. Genetic drift is most pronounced in small populations, where genotypes with a neutral and even deleterious effect have a high chance of surviving, which fosters ITH [177,178]. It is along these lines of reasoning that the “big bang” model of colorectal cancer carcinogenesis was formulated [179]. According to big bang carcinogenesis, all measurable genetic variation emerges immediately after transformation when the population size is small.

Confinement of malignant cells into spatial structures has the effect of solid malignancies acting as if they are collections of small populations, again fostering ITH [178]. This compartmentalization of competing populations becomes strikingly clear from the dynamics of xenografted cancers and bacterial colonies, which grow primarily on the surface [180,181]. Simulations of tumor growth demonstrate that the number of cells in competition in each compartment has a dramatic impact on ITH [182]. The spatial structure of a tumor largely derives from the tissue of origin, and variations in ITH between cancer types might be partially explained by their different tissue architecture [182,183,184]. Motility increases the number of competitors of cells and allows cells with a fitness benefit to expand rapidly and reduce ITH [185,186]. The difference in cell motility between hematological and non-hematological cancers might explain the fact that despite high CIN, karyotype heterogeneity is relatively low in hematological cancers [4].

## 6. Conclusions

Karyotype ITH is a prevalent characteristic of cancers derived from all tissues. The level of ITH is a predictor of cancer progression and patient survival. Understanding why cancers reach different levels of karyotype ITH is, hence, of interest for fundamental tumor biology and translational research. Here, we aimed to provide an overview of the factors that contribute to karyotype ITH. Karyotype ITH is generated by CIN. The karyotype is an important determinant of cell phenotype and is thereby under selection from cell intrinsic and cell extrinsic pressures. Under stringent selection, only a few karyotypes have compatible fitness and karyotype ITH will, hence, be limited.

Karyotype ITH is further limited through the competition of malignant cells. Cell completion can act directly, with cells modulating the fitness of surrounding cells, or indirectly as a competition for available space and resources. Random cell death can eliminate cells with a fit karyotype from the malignant cell population. On the other hand, substantial ITH can persist if competition between malignant cells is restrained by the tissue architecture and lack of cell migration. Karyotype ITH is, hence, a complex phenomenon, emerging from CIN in a context-specific manner that depends on the ecological characteristics of a cancer, including its geometric structure.

## Figures and Tables

**Figure 1 cancers-14-04986-f001:**
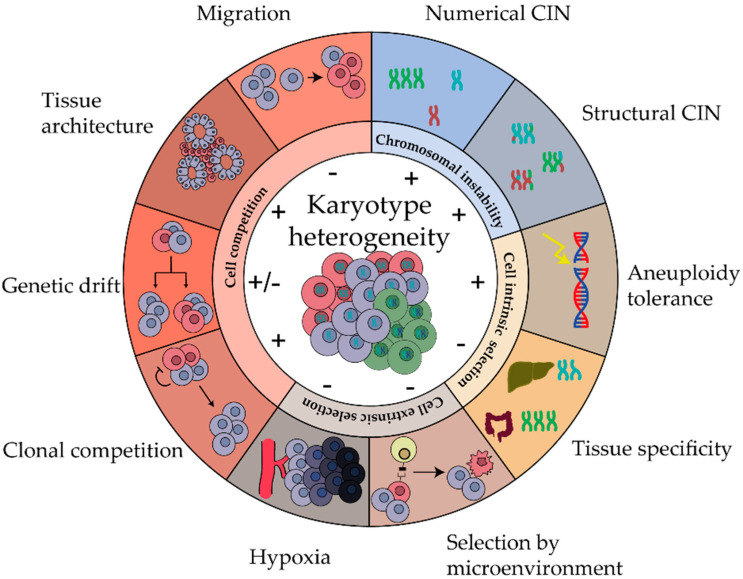
Biological processes and properties that shape karyotype heterogeneity. Karyotype heterogeneity is increased or decreased by chromosomal instability, cell intrinsic and extrinsic selection and cell competition.

**Figure 2 cancers-14-04986-f002:**
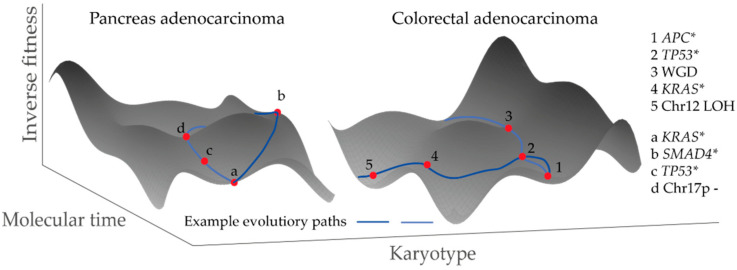
Illustrations of karyotype fitness landscapes and how they evolve during carcinogenesis, in analogy to the cell differentiation landscapes as proposed by Waddington [114]. Examples of evolutionary paths are highlighted. * Mutation of the named gene.

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
