# Peer review of "Chromosomal Instability, Selection and Competition: Factors That Shape the Level of Karyotype Intra-Tumor Heterogeneity"

_cancers, 2022, doi:10.3390/cancers14204986_

Round 1
Reviewer 1 Report
In the manuscript “Chromosomal instability, selection and competition: factors that shape the level of karyotype intra-tumor heterogeneity” authors review cellular processes and ecological properties that determine intra-tumor heterogeneity (ITH) of karyotypes. Since the level of ITH is a predictor of cancer progression and patient survival, understanding why cancers reach different levels of karyotype ITH is of great interest for fundamental tumor biology and translational research. Authors aimed to provide an overview of the factors contributing to karyotype ITH, namely, chromosomal instability as a source of karyotypic heterogeneity, and factors that shape ITH: cell intrinsic and extrinsic pressures, competition and collaboration inside the cancer cell population, immune-microenvironment, and the implications all these factors have for the subsequent selection for karyotypes.
The Review is well written and organized, with the title adequate for the content. It has 173 relevant references and two Figures.
Sections 3-5 describe intrinsic and extrinsic selective pressures, role of microenvironment, immunoediting, role of hypoxia and competition between malignant cells, and provide a broad and comprehensive framework for understanding factors that shape ITH.
Section 2, “Numerical and structural CIN generates new karyotypes”, offers relatively short and limited overview of mechanisms that lead to chromosomal instability in cancer cells.
This Review is a significant contribution to the field of tumor biology and evolution of cancer cells and can be published after minor changes.
Figure 2 requires more detailed explanation in the text or Figure legend.
L257: “immunocompetent” instead of “immunocomponent” and “immunogenic” instead of “immunogenetic”.
L258: “immune pressure” instead of “immunogenic pressure”.
L260-261: the sentence needs editing.
L290-296: when the citation of Santaguida et al. [131] is absolutely appropriate at the beginning of the paragraph, it is not the right citation for the last sentence: “One of the best described proinflammatory pathways is the cGAS-STING pathway [131]”
1
Author Response
In the manuscript “Chromosomal instability, selection and competition: factors that shape the level of karyotype intra-tumor heterogeneity” authors review cellular processes and ecological properties that determine intra-tumor heterogeneity (ITH) of karyotypes. Since the level of ITH is a predictor of cancer progression and patient survival, understanding why cancers reach different levels of karyotype ITH is of great interest for fundamental tumor biology and translational research. Authors aimed to provide an overview of the factors contributing to karyotype ITH, namely, chromosomal instability as a source of karyotypic heterogeneity, and factors that shape ITH: cell intrinsic and extrinsic pressures, competition and collaboration inside the cancer cell population, immune-microenvironment, and the implications all these factors have for the subsequent selection for karyotypes.
The Review is well written and organized, with the title adequate for the content. It has 173 relevant references and two Figures.
Sections 3-5 describe intrinsic and extrinsic selective pressures, role of microenvironment, immunoediting, role of hypoxia and competition between malignant cells, and provide a broad and comprehensive framework for understanding factors that shape ITH.
Section 2, “Numerical and structural CIN generates new karyotypes”, offers relatively short and limited overview of mechanisms that lead to chromosomal instability in cancer cells.
This Review is a significant contribution to the field of tumor biology and evolution of cancer cells and can be published after minor changes.
We thank the reviewer for the positive evaluation of our work. We have expanded the discussion on numerical and structural CIN in section 2, which was indeed relatively short compared to the other sections of this manuscript.
Figure 2 requires more detailed explanation in the text or Figure legend.
We have expanded the description of this figure in both the main text and the figure legend. In particular, we explain the analogy of this figure to the Waddington fitness landscape of epigenetic changes for cell differentiation (Waddington, The strategy of genes 1957). The accumulated genetic aberrations can similarly be viewed as “differentiation” of cancer (cells) resulting in fitness landscape of karyotypes.
L257: “immunocompetent” instead of “immunocomponent” and “immunogenic” instead of “immunogenetic”.
We thank the reviewer for pointing this out and have adjusted the text.
L258: “immune pressure” instead of “immunogenic pressure”.
We thank the reviewer for pointing this out and have adjusted the text.
L260-261: the sentence needs editing.
We have split-up this sentence to make the text more clear.
L290-296: when the citation of Santaguida et al. [131] is absolutely appropriate at the beginning of the paragraph, it is not the right citation for the last sentence: “One of the best described proinflammatory pathways is the cGAS-STING pathway [131]”
We agree, reference was 131 was not the most appropriate at the end of this paragraph. We replaced it with a reference to a recent review on cGAS-STING in cancer by Vashi and Bakhoum (Vashi and Bakhoum, Trends in biomedical sciences 2021).
Reviewer 2 Report
In this manuscript van der Bosch, Derks and Miedema describe the factors that contribute to large variability of karyotype ITH. They displayed the role of chromosomal instability in tumors and karyotype ITH determination. Finally, they point out the competitive behavior of malignant cells that hence limits ITH. They discuss also immune-microenvironment of cancer and its effect on karyotype divergence.
Overall evaluation:
The manuscript contains few sections. Since section 4 and 5 is very clear and contains many comprehensive information concerning CIN, cellular fitness, role of microenvironment and cell competition, the previous sections require some precisions.
Important aspect that considers silenced cancer stem cells has not been mentioned in the manuscript. Those cells might have limited CIN and hence they are often responsible for tumor resistance or/and metastatic progression. May authors comment on that?
Other specific concerns and comments:
Section 2 lines: 97-101 needs more information, what is the conclusion of referenced works?
Section 3.1 line 123: others traits than adiposity should be mentioned
Section 3.1 lines: 139-140: “Although aneuploidy is a common characteristic in cancers, homozygous deletions are rarely found” this sentence needs explanation, why homozygous deletions are rare in cancers?
Section 3.2 lines: 155-157 “The impact of aneuploidy on gene expression is also evident from the fact that copy number profiles can be inferred from measurements of gene expression of bulk samples [74] and single-cells [75,76].” May authors precise what is the rapport of this sentence with the rest of section?
Section 3.2 line 167 : “this link between karyotype and phenotypic ITH provides an explanation” Authors do not provide enough explanation in this section between genomic and phenotypic ITH, the conclusion sentence needs to be revised.
Section 3.3 May authors comment on the relation in terms of ITH between primary tumor and its metastasis, does the tissue origin matters or seeding site may shape ITH?
Section 3.4 lines: 200-201, statement: “either mutated or amplified“ is doubled in this sentence. Would be appreciated to mention here how the karyotype is shaped by KRAS?
Author Response
In this manuscript van der Bosch, Derks and Miedema describe the factors that contribute to large variability of karyotype ITH. They displayed the role of chromosomal instability in tumors and karyotype ITH determination. Finally, they point out the competitive behavior of malignant cells that hence limits ITH. They discuss also immune-microenvironment of cancer and its effect on karyotype divergence.
Overall evaluation:
The manuscript contains few sections. Since section 4 and 5 is very clear and contains many comprehensive information concerning CIN, cellular fitness, role of microenvironment and cell competition, the previous sections require some precisions.
We thank the reviewer for this overall positive evaluation of our work. In line with the comments below, and the comments of reviewer #1, we have expanded and improved section 2 and 3.
Important aspect that considers silenced cancer stem cells has not been mentioned in the manuscript. Those cells might have limited CIN and hence they are often responsible for tumor resistance or/and metastatic progression. May authors comment on that?
Cancer stem cells are thought to have lower genomic instability (Van Wely, OncoImmunology 2012). Or at least, less CIN than non-cancer stem cells (Godek, Cancer Discovery 2016). We added a discussion on cancer stem cells and CIN to section 3.1.
Other specific concerns and comments:
Section 2 lines: 97-101 needs more information, what is the conclusion of referenced works?
These studies linked specific CNV signatures to the underlying biological processes, such as homologous recombination deficiency. Interestingly, not for all CNV signatures the etiology could yet be explained, suggesting unknown biological process that drive distinct types of CIN. We added this more detailed information on CNV signatures to section 2.
Section 3.1 line 123: others traits than adiposity should be mentioned
We added liver damage and decreased intelligence.
Section 3.1 lines: 139-140: “Although aneuploidy is a common characteristic in cancers, homozygous deletions are rarely found” this sentence needs explanation, why homozygous deletions are rare in cancers?
We move the observation that homozygous deletions are rarely found in cancers, which was done in ref 73 and 74, to section 3.2 where the impact of aneuploidy on cell fitness is discussed. We speculate that homozygous deletions of large genomic segments are rarely found because they eliminate genes from a cell with a negative consequence on cell fitness.
Section 3.2 lines: 155-157 “The impact of aneuploidy on gene expression is also evident from the fact that copy number profiles can be inferred from measurements of gene expression of bulk samples [74] and single-cells [75,76].” May authors precise what is the rapport of this sentence with the rest of section?
The idea here is that if you can measure karyotypes from gene expression profiles, the impact of karyotypes on gene expression is significant (otherwise it would not be possible to infer karyotypes). We consider this inverse relation a relevant illustration of the relation between karyotypes and gene expression. We have adjusted the text to make this point more clear in the revised manuscript.
Section 3.2 line 167 : “this link between karyotype and phenotypic ITH provides an explanation” Authors do not provide enough explanation in this section between genomic and phenotypic ITH, the conclusion sentence needs to be revised.
We thank the reviewer for pointing this out and agree that no solid link between karyotype and phenotype is given in this section. What is given is solid evidence for the impact of karyotypes on gene expression. We rewrote the concluding paragraph of this section accordingly.
Section 3.3 May authors comment on the relation in terms of ITH between primary tumor and its metastasis, does the tissue origin matters or seeding site may shape ITH?
We thank the reviewer for suggesting this interesting point. It appears that CNVs do change between primary tumors and metastases. However, the level of heterogeneity between patients is consistently higher than within a patient for osteosarcoma (Wang, Cancer Research 2019). Moreover, no consistent pattern of CNV changes is found for CRC metastasis (Mamlouk, Nature Communications 2017). Also the level of karyotype ITH was found to be similar within primary CRCs compared to peritoneal metestasis (Lenos, Nat Communications 2022). This observation supports the hard wiring of gene expression discussed in section 3.3, to which we added the above.
Section 3.4 lines: 200-201, statement: “either mutated or amplified“ is doubled in this sentence. Would be appreciated to mention here how the karyotype is shaped by KRAS
The idea here is that KRAS mutations do not coalesce with KRAS amplification. A mutation in KRAS hence negatively associates with amplification of the same gene. We have rewritten the text in the revised manuscript to make this more clear.